# Raman time-delay in attosecond transient absorption of strong-field created krypton vacancy

Li Wang[1,2,3], Guangru Bai[1,2,3], Xiaowei Wang [1,2] ✉, Jing Zhao [1,2] ✉, Cheng Gao [1,2], Jiacan Wang[1,2], Fan Xiao[1,2], Wenkai Tao[1,2], Pan Song[1], Qianyu Qiu[1,2], Jinlei Liu [1,2] & Zengxiu Zhao [1,2] ✉

Strong field ionization injects a transient vacancy in the atom which is entangled to the outgoing photoelectron. When the electron is finally detached, the ion is populated at different excited states with part of coherence information lost. The preserved coherence of matter after interacting with intense short pulses has important consequences on the subsequent nonequilibrium evolution and energy relaxation. Here we employ attosecond transient absorption spectroscopy to measure the time-delay of resonant transitions of krypton vacancy during their creation. We have observed that the absorptions by the two spin-orbit split states are modulated at different paces when varying the time-delay between the near-infrared pumping pulse and the attosecond probing pulse. It is shown that the coupling of the ions with the remaining field leads to a suppression of ionic coherence. Comparison between theory and experiments uncovers that coherent Raman coupling induces time-delay between the resonant absorptions, which provides insight into laser-ion interactions enriching attosecond chronoscopy.

Generation of coherent extreme ultraviolet (XUV) sources such as isolated attosecond pulses (IAP) enables the probing of electron dynamics beyond chemical reactions, deep into the atomic core for various phases of matter at attosecond resolution[1–4]. The emerging attosecond science has been found impactful on condensed matter physics, material science, chemistry, and biology. Particularly, attosecond chronoscopy has been established where the time or time-delay become the new observables[5–7]. For example, the time-delay difference of photoemission from the 2s and 2p orbitals of Ne has been measured as 20 attoseconds[8], revealing that less time is required for the tighter bounded electrons to escape. Similar investigation on molecules demonstrates that the measured time-delay provides the information of shape resonance which is directly related to the details of the atomic potential[9]. In liquids[10], it is shown that the multiple scattering by the environment during the electron

escaping does not change the local time delay between different ionization channels.

The concept of time-delay can be traced back to Wigner[11]. The Wigner time-delay is originally referred to the additional time required when the electron wave packet is scattered by the atomic potential. Comparing to the field-free motion, each continuum state experiences a scattering phase shift and the time-delay is therefore defined as the derivative of the phase shift with respect to energy. The Wigner time-delay is indeed the other facet of the quantum coherence of the wave packet by which counting the scattering phase variation is meaningful. It can be generalized to other processes whenever a continuum is involved, providing a fresh look at various fundamental processes in time domain, such as tunneling ionization[12–14], photoemission[8,9], Auger decay[15], electron correlation[16], AC Stark shift[17], spin-orbit coupling[1], electron

[1]Department of Physics, National University of Defense Technology, Changsha 410073, China. [2]Hunan Key Laboratory of Extreme Matter and Applications, National University of Defense Technology, Changsha 410073, China. [3]These authors contributed equally: Li Wang, Guangru Bai. ✉e-mail: xiaowei.wang@nudt.edu.cn; jzhao@nudt.edu.cn; zhaozengxiu@nudt.edu.cn

transportation[18] and recollision dynamics[19]. So far, less attention has been paid to the time-delay of resonant absorption. According to quantum mechanics, bound states are discrete states with infinitesimal level width when level broadening is ignored. The resonant absorption of light thus invalids the argument of the Wigner time-delay defined on the dispersion of continuum electron wave packet[11]. However, when the bound states are coupled by a laser field, the time-delay can still be properly defined and measured as will be demonstrated.

With one of the 4p electrons removed, the strong field ionization of krypton (Kr) creates two spin-orbit split ionic states that form a spin-orbit wave packet evolving in the remaining laser field[1]. When the electron is finally detached, the ion is left in a statistical population of different excited states with part of the coherence information lost. Utilizing an IAP, it is possible to probe the resonant absorptions of the transient created ions in time domain. As quantum coherence measures the spatiotemporal correlations of the dynamics, the time spent by the absorption is thus directly related to the coherent evolution of the ionic wave packet[20–24]. While previous pioneering work has partially characterized the ionization dynamics and the created electron wave packet[1,21,25], here we fully survey the whole process starting from the transient ionization to the ion-laser coupling and the attosecond XUV absorption.

In this work, we apply an IAP to probe the transient absorption by Kr vacancy during its creation in a near-infrared (NIR) strong laser field from the neutral. By varying the time-delay between the NIR field and the IAP, we have observed that the two resonant absorptions are modulating at different paces, despite the field-free energy separation of the two absorption lines is as narrow as 0.6 eV. It demonstrates that the quantum coherences of both the tunneling electron and the injected ion play key roles in the transient absorption. We have found that the coupling of the coherent populated ion with the laser field initiates a coherent Raman process causing the time-delay between the absorption paths. It indicates that matter can preserve part of the coherence after interacting with intense short pulses, which is crucial for the understanding of nonequilibrium evolution and energy relaxation occurring in laser-matter interactions such as laser-ablation and laser-machining.

## Results

### Attosecond chronoscopy

The attosecond transient absorption spectroscopy (ATAS) as illustrated in Fig. 1a was employed to measure the time-delay of electron transition dynamics of Kr. A NIR laser beam with pulse duration 5.3 fs, center wavelength about 730 nm, and pulse energy 1.8 mJ [see supplementary materials (SM) note 1 for more details] was divided equally into two parts by a 50:50 beam splitter. One arm passed through the double optical gating[26] optics and was focused into neon gas-filled cell by a concave mirror with focal length of 350 mm to generate IAP. A 200 nm thick Zr foil was used to filter out residual NIR beam, then a gold coated toroidal mirror focused the IAP into the second gas cell filled with Kr samples. The other arm which transmitted through the beam splitter propagated along a delay line and combined with IAP by a hole-drilled mirror. The pump beam was focused by a lens with focal length of 400 mm, to generate vacancy in Kr valence shell. The IAP served as the probe of the vacancy generation dynamics. After interacting with Kr gas, the residual NIR was filtered out again using a 200 nm thick Zr foil, and the transmitted IAP was recorded by an in-situ calibrated (see SM note 2) home-made XUV spectrometer consisted of a flat field grating and a microchannel plate (MCP) detector. The time delay between the NIR and the IAP was controlled with active phase locking technology[27], and the time jitter and scan step size were 23.8 as and 141 as, respectively, which allowed for resolving periodical dynamics with time resolution as high as 8 as (see SM note 3). The measured time-resolved change of optical density ($\Delta OD$) is showcased in Fig. 1c. The $\Delta OD$ is defined as variation of optical density of the Kr$^+$ ions with respect to that of Kr atoms: $\triangle OD = -\log\frac{S_x^i}{S_x^0} + \log\frac{S_x^n}{S_x^0} = -\log\frac{S_x^i}{S_x^n}$, where $S_x^i$ and $S_x^n$ are the IAP spectral intensity after the absorption by ionic Kr$^+$ and neutral Kr, respectively, and $S_x^0$ is the unabsorbed spectral intensity.

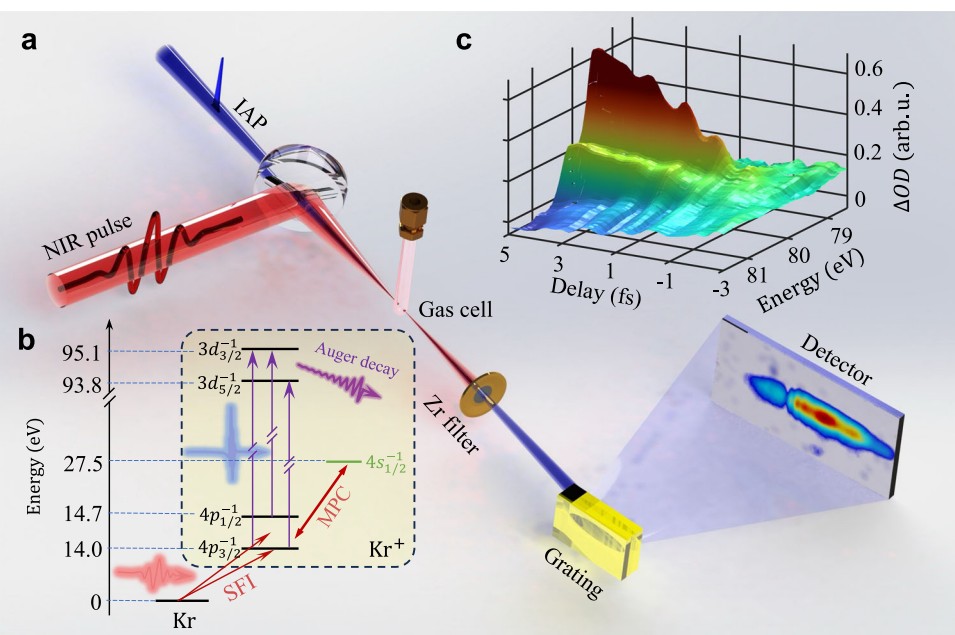

**Fig. 1 | Illustration of attosecond transient absorption spectroscopy scheme.** The strong field ionization (SFI) of Kr injects the ion in the two spin-split states which are multi-photon coupled (MPC) to the neighboring state (4 s) of Kr$^+$. The isolated attosecond pulse (IAP) probes the resonant absorption of the ion by exciting the electron into 3d hole state which is relaxing through Auger processes, (**a**) experimental setup; (**b**) the level diagram and involved dynamics and (**c**) time-resolved transient change of optical density.

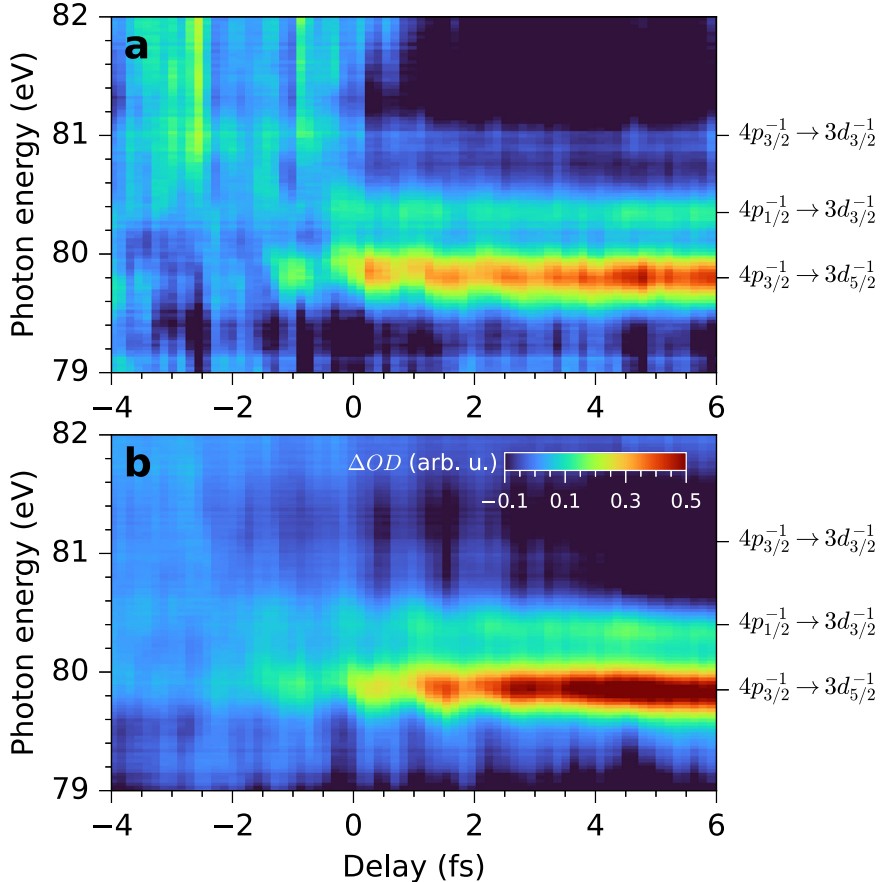

**Fig. 2 | Time-resolved absorption spectra at two pump laser intensities.** The change of optical density was measured with laser intensities of (**a**) $I_1 = 3.6(\pm 0.3) \times 10^{14}$ W/cm² and (**b**) $I_2 = 4.6(\pm 0.3) \times 10^{14}$ W/cm². Negative delay indicates that the IAP arrives before the NIR.

## Time-resolved absorption spectra

Figure 2 presents the smoothed (see SM note 4 for raw data) energy- and time-resolved $\Delta OD$ of Kr valence vacancy at two laser intensities of $I_1 = 3.6(\pm 0.3) \times 10^{14}$ W/cm² (Fig. 2a) and $I_2 = 4.6(\pm 0.3) \times 10^{14}$ W/cm² (Fig. 2b) with the corresponding Keldysh parameters[28] of $\gamma \approx 0.55$ and 0.62, respectively. It can be seen that the strongest absorption line, i.e. the transition from $4p_{3/2}^{-1}$ to $3d_{5/2}^{-1}$ (79.8 eV) is more sensitive to the NIR intensity, which is consistent with intensity-dependent ionization rate predicted by tunneling ionization theory[29,30]. Two additional absorption lines on the energy axis from low to high correspond to the transitions of $4p_{1/2}^{-1} \rightarrow 3d_{3/2}^{-1}$ (80.4 eV) and $4p_{3/2}^{-1} \rightarrow 3d_{3/2}^{-1}$ (81.1 eV), respectively. We focus on the former two lines as the third is the weakest and was already discussed previously[1]. The delay dependent line shape deformation in both spectra has been attributed to time dependent dipole phase change induced by the residual laser field[1,20].

The measured delay dependent $\Delta OD$ for the two concerned transitions are depicted as scatters in Fig. 3a, b for NIR laser intensity of $I_1$ and $I_2$, respectively. It can be seen $\Delta OD$ shows an overall increase during the presence of the NIR laser field. Furthermore, the increasing is not monotone but modulating with a period of half optical cycle of the laser field[1], to visualize which the experimental data points are fitted by oscillational curves (thin solid lines) as shown in the figures. The increase of $\Delta OD$ can be attributed to the accumulated ionic population, while the modulations need more careful investigations. In particular, the two absorption channels reach local maximum/minimum $\Delta OD$ at different delay, suggesting a time-delay difference between the two absorptions. The measurements show that the temporal oscillation of $\Delta OD$ for $4p_{3/2}^{-1} \rightarrow 3d_{5/2}^{-1}$ lags that for $4p_{1/2}^{-1} \rightarrow 3d_{3/2}^{-1}$

by 420 and 480 attoseconds at $I_1$ and $I_2$ respectively. Similar phenomenon observed in xenon atoms was previously reported but not explained[25].

## Discussion
### Theoretical model

To uncover the involved dynamics, the physics behind the time-delay of the two resonant transitions are investigated by adopting density matrix approach[31] (see Methods and SM note 5) with the relevant states included:

$$\frac{d\rho^+}{dt} = -\frac{i}{\hbar}\left[H^+, \rho^+\right] + \left[\dot{\rho}^+\right]_{ion} + \left[\dot{\rho}^+\right]_{decay} \quad (1)$$

where $\rho^+$ denotes the density matrix of the ion, and the second (third) term on the right represents the injection (decay) of ion. This model has been successfully applied to the study of coherent emission from nitrogen ion[32], here it is further developed to account for transient absorption. The following processes are considered as illustrated in Fig. 1b: (i) the ionization of the neutral under strong field approximation (SFA)[24]; (ii) the coupling of the two ionic states $4p_{3/2}^{-1}$ and $4p_{1/2}^{-1}$ (only $m_j = -1/2$ considered) with the neighboring 4 s configurations of the ion due to the NIR pulse; (iii) transitions from 4p orbitals to 3d orbitals induced by the IAP polarized in parallel with the NIR pulse; (iv) the decay of the 3d hole states with lifetime of 7.5 fs due to Auger processes. All the states are evolving under both the NIR and IAP pulses with the time-dependent Hamiltonian $H^+(t)$. The absorption spectra are calculated from the dipole response to the combined field of both pulses at given time delays.

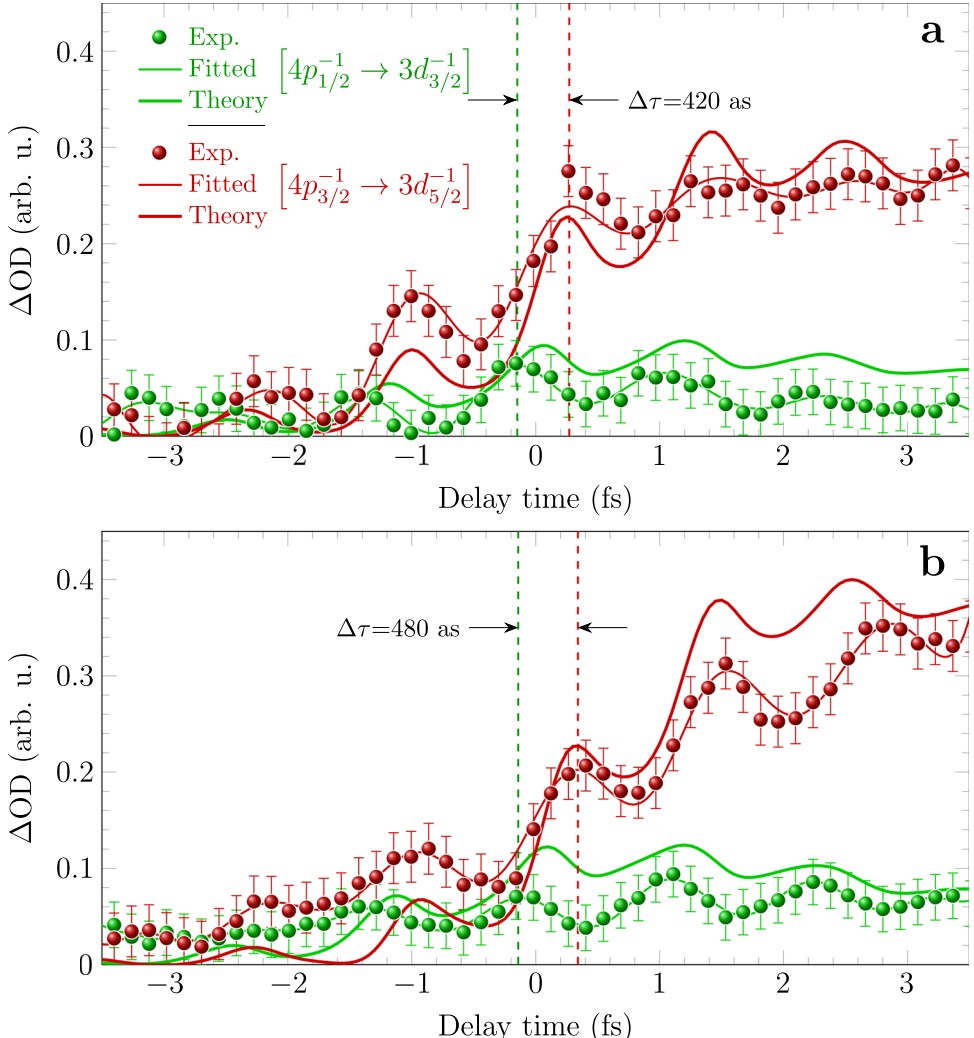

**Fig. 3 | Delay-dependent ΔOD for transitions $4p_{3/2}^{-1} \to 3d_{5/2}^{-1}$ and $4p_{1/2}^{-1} \to 3d_{3/2}^{-1}$.** The ΔOD for transitions $4p_{3/2}^{-1} \to 3d_{5/2}^{-1}$ (red) and $4p_{1/2}^{-1} \to 3d_{3/2}^{-1}$ (green) ramp up oscillatory under pump laser intensities of $I_1$ (**a**) and $I_2$ (**b**). The experimental data points (scatters) are fitted with smooth oscillational curves (thin solid curves). The theoretical predicted ΔOD (thick solid curves) based on density matrix approach reproduce the experimental results qualitatively. The dashed lines indicate the local maximum of the fitted curves, suggesting time difference of 420 attoseconds and 480 attoseconds for the two transitions with NIR laser intensity of $I_1$ and $I_2$, respectively. The error bars show the uncertainty of ΔOD based on the standard deviation of measure spectra (see SM note 4).

## Coherence of ionization

The full simulation reproduces the experimental results qualitatively as shown in Fig. 3. The calculated ΔOD increases with the elapsed time for both transitions as expected. In particular, the oscillation nature of ΔOD, which has been attributed to the polarization of the neutral ground state induced reversible ionization[25], is well-reproduced by the theory. Here we employ the ionization model based on SFA[23,24] which takes into account the coherence of the escaped photoelectron wave packet. It is found that the obtained ionization probabilities for laser intensities of $I_1$ and $I_2$ exhibit oscillations over time as indicated by the black curves in Fig. 4a, d respectively, in contrast to stair-like increasing of ionization probability calculated with ADK model[29] (see SM note 6 for details). The reason is that the ADK model assumes ionization is finished instantaneously and the quantum coherence between the continuum and the ground state is lost right after tunneling. Our study suggests the coherence of ionization persists and should be considered in addition to the ionization rate even for single active electron system. Once the SFA ionization rate is obtained, the injection rates of the two spin-orbit states $4p_{3/2}^{-1}$ and $4p_{1/2}^{-1}$ are weighted by 2/3 and 1/3 respectively. For the coherence of the two states of the remaining

ions, we assume the instantaneous ionization creates maximum coherence among them referred as coherent injection. For comparison, the case of incoherent injection is simulated as well by assuming that the off-diagonal element of the two-state density matrix are zero during the ionization injection.

## Coherence-driven population transfer

The populated ionic states can be coupled with other states by the strong laser field, resulting in modified occupation and coherence. Note that the two spin-orbit split states can be coupled to $4s^{-1}$ state resonantly with transition energy close to 9-NIR-photon. Those states constitute a Λ-type three-level system. The population is thus transferred between the two lower states through Raman process. However, we notice that the direction of population transfer is modified by the injected coherence of the ion. When the two ionic states are populated incoherently, the population of $4p_{3/2}^{-1}$ ($\rho_{11}$) is reduced and that of $4p_{1/2}^{-1}$ ($\rho_{22}$) is increased, indicating population transfer from $4p_{3/2}^{-1}$ to $4p_{1/2}^{-1}$ (see the green curves). On the contrary, with coherent injection, we found surprisingly that the direction of population transfer is reversed as shown in Fig. 4a, d. This coherence-driven population transfer leads to increased population

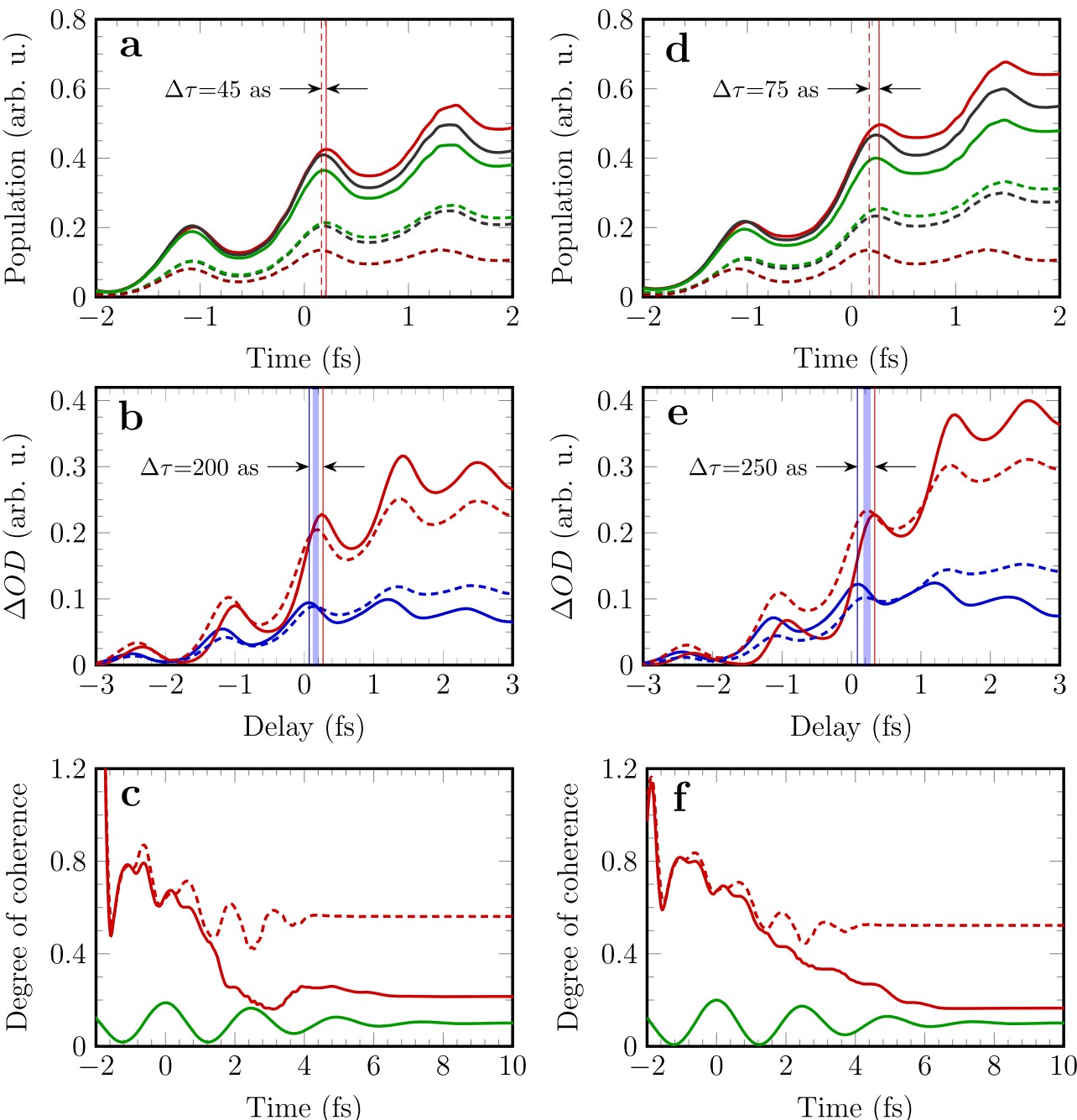

**Fig. 4 | Evolution of ionic population, change of optical density and coherence at the laser intensity $I_1$ (a,b,c) and $I_2$ (d,e,f). a, d** Time-dependent population of the two ionic states $4p_{3/2}^{-1}$ (solid lines) and $4p_{1/2}^{-1}$ (dashed lines) calculated for coherent (red) and incoherent injection (green). Black curves are the population calculated by SFA model; **b, e** $\Delta OD$ for transitions $4p_{3/2}^{-1} \rightarrow 3d_{5/2}^{-1}$ (red) and $4p_{1/}$ $_2^{-1} \rightarrow 3d_{3/2}^{-1}$ (blue) with (solid) and without (dashed) considering the coupling to the neighboring state $4s_{1/2}^{-1}$; **c, f** degrees of coherence calculated with (solid lines) and without (dashed lines) the Raman coupling. The waveforms of the NIR pulse are also plotted (green lines).

on the $4p_{3/2}^{-1}$ state, which enhances its absorbance as observed in the experiments (see Fig. 3a, b).

**Raman time-delay and the ionic coherence**
The time-delay difference between the two transitions can also be understood by considering the Raman process. Firstly, as indicted by coherent injection model shown as red lines in Fig. 4a, d, the time-delays between the two population modulations are only 45 and 75 attoseconds at the two laser intensities respectively, which are too small comparing to the observed time difference in $\Delta OD$ traces. The calculated $\Delta OD$ for transitions $4p_{3/2}^{-1} \rightarrow 3d_{5/2}^{-1}$ (red) and $4p_{1/2}^{-1} \rightarrow 3d_{3/2}^{-1}$

(blue) with (solid lines) and without (dashed lines) coupling to $4s_{1/2}^{-1}$ state are shown in Fig. 4b, e [see SM note 7 for details]. It can be seen the coupling with 4s state causes the reduction of the $4p_{1/2}^{-1}$ absorption and enhancement of the $4p_{3/2}^{-1}$ absorption at both laser intensities, which reflects the transfer of population. Moreover, the Raman coupling induces an extra phase delay for $4p_{3/2}^{-1} \rightarrow 3d_{5/2}^{-1}$ and phase advance for $4p_{1/2}^{-1} \rightarrow 3d_{3/2}^{-1}$. Without coupling with $4s_{1/2}^{-1}$, the calculation gives imperceptible time difference, as indicated by the vertical rectangular in Fig.4b, e. On the contrary, the Raman process comes into play, and the time-delay between the two transitions is calculated to be 200 as for $I_1$ (Fig. 4b) and 250 as for $I_2$ (Fig. 4e). Although both

results differ from the experimental observations by about a factor of two, they do trace the large time difference between different transition channels. With more ionic states involved in the Raman process, the results would get improved slightly [see SM note 8]. Both the population and the phase shift of the dipole response are found affected by the Raman coupling [see Fig. S13 in SM], which unambiguously confirms the Raman time delay. Additional time delay may be addressed by taking into account the core-excited virtual states of the neutral[33], or the non-adiabatic ionization effect[25].

Finally, we consider the coherence of the two states, i.e., the non-diagonal term of the density matrix. The degree of coherence can be defined as $g = |\rho_{12}|/\sqrt{\rho_{11}\rho_{22}}$[1], where $\rho_{12}$ is the non-diagonal element of the density matrix, and $\rho_{11}$ and $\rho_{22}$ are the population of $4p_{3/2}^{-1}$ and $4p_{1/2}^{-1}$ shown in Fig. 4a, d. The evolution of ionic coherence is shown in Fig. 4c, f for the two laser intensities. In case that the Raman coupling is absent, the coherence of the ion is purely accumulated by the continuous coherent injection of ions as the ion-laser interaction is shut down. It can be seen in Fig. 4c that the degree of coherence increases with time and reaches maximum before the laser reaches peak field strength at time-zero. After that the coherence starts to fall down and eventually reaches a constant value of 0.56 after the vanishing of the laser pulse at intensity of $I_1$. For the higher intensity $I_2$, the same behavior is observed except that the final degree of coherence is smaller (0.52). It hints that ions are produced with less coherence for higher intensity and longer pulse duraitons[34]. The decrease of coherence while the ion population increases can be rationalized by realizing that the coherence injected via ionization at different optical cycle is not in phase or synchronized with the transient ionization. When the Raman coupling steps in, the degrees of coherence drop at both laser intensities shown in red curves in Fig. 4c, f, indicating that the Raman coupling induces extra phase shift of the coherence in addition to the modification of the magnitude via population transfer. The residual coherence after the pumping pulse has important consequences for the observation of the valence electron motion of the cation and the reconstruction of the density matrix[20].

In conclusion, we report the resolving of the evolution of Kr from neutral to ions under the strong driving laser field with ATAS. We observe the modulation of change of optical density for the two spin-orbit split states of the cation and find they differ in time with a few hundred attoseconds. By developing a comprehensive theory to include all the relevant dynamics, we identify the crucial roles of coherence of ionization and the resulted ionic coherence. We demonstrate that the continued interaction of the ion with the remaining laser field modifies the population distribution as well as their coherence among the vacancy states. It is shown the coherent Raman process leads to the observed time-delay of the two resonant absorptions. The concept of Raman-delay provides insight into laser-dressed bound-bound transitions as well as transient ionization injection enriching attosecond chronoscopy.

## Methods
### Experimental apparatus
The laser system used in experiments was a 10-pass chirped pulse amplification system (Femtopower HE) operated at 1 kHz. It delivered 25 fs, 4.2 mJ pulses centered at 800 nm. The multi-cycle pulses were then coupled to a 1 m long helium-filled hollow-core fiber with diameter of 300 $\mu m$. To decrease the ionization effect and improve the throughput of the fiber, differential pumping scheme was employed. High pressure helium gas (2000 mbar) was introduced from the exit of the fiber, while pressure of the entrance was kept as low as 3E-2 mbar with a dry pump (TriScroll 300). The spectrum was then broadened to cover a spectral range of 580–940 nm (see Fig. S2 in SM for details). To get ultrashort few-cycle pulses, chirped mirrors (Laser Quantum DCM7) were used for group delay dispersion

compensation. After the chirped mirrors, the NIR pulse was compressed down to 5.3 fs, which was measured with a fringe resolved autocorrelator (Femtometer). The FWHM of the autocorrelation signal is 10 fs, so the duration of NIR pulses is estimated to be 5.3 fs assuming sech2 temporal shape. The pointing stability of the NIR laser pulses was measured to be better than 20 $\mu rad$. The power stability after hollow-core fiber was measured with a power-meter (Thorlabs PM100D), and the root-mean-square (rms) error of the laser power was estimated to be 1.1%. The ATAS system is introduced in the main text and explained in detail in the SM note 1–4 of supplementary materials.

### Numerical calculation
To calculate the transient absorption spectra of Kr⁺, we numerically solve the strong field ionization-coupling equation Eq. (1) with the time-dependent ionic Hamiltonian given as:

$$H^+ = \begin{pmatrix} E_1 & 0 & -\varepsilon(t)d_{13} & -\varepsilon(t)d_{14} & -\varepsilon(t)d_{15} \\ 0 & E_2 & -\varepsilon(t)d_{23} & 0 & -\varepsilon(t)d_{25} \\ -\varepsilon(t)d_{13} & -\varepsilon(t)d_{23} & E_3 & 0 & 0 \\ -\varepsilon(t)d_{14} & 0 & 0 & E_4 & 0 \\ -\varepsilon(t)d_{15} & -\varepsilon(t)d_{25} & 0 & 0 & E_5 \end{pmatrix} \quad (2)$$

which can be abbreviated as $H_0^+ - \mu\varepsilon(t)$ with $\mu$ being the transition dipole matrix. The five states are numbered according to their energy in ascending order. The field $\varepsilon(t)$ including both the NIR pulse and the delayed attosecond XUV pulse. The widths of the NIR pulse and the IAP are given by $\tau_{NIR} = 5.3$ fs and $\tau_{XUV} = 150$ attoseconds. The laser frequency is $\omega_{IR} = 1.65$ eV, and the central photon energy of the XUV pulse is $\omega_{XUV} = 85$ eV.

The ionization can leave the Kr⁺ populated on either $4p_{3/2}^{-1}$ or $4p_{1/2}^{-1}$. Assuming they are coherently populated with the injected state as

$$|l=1, m_l=0, s=-1/2\rangle = \sqrt{\tfrac{2}{3}}\left|4p_{\frac{3}{2}}^{-1}\right\rangle + \sqrt{\tfrac{1}{3}}\left|4p_{\frac{1}{2}}^{-1}\right\rangle, \quad (3)$$

and the matrix form of $\left[\dot{\rho}^+\right]_{ion}$ can be written as

$$\left[\dot{\rho}^+\right]_{ion} = \begin{pmatrix} \frac{2}{3}W(t)\rho_0 & \frac{\sqrt{2}}{3}W(t)\rho_0 & 0 & 0 & 0 \\ \frac{\sqrt{2}}{3}W(t)\rho_0 & \frac{1}{3}W(t)\rho_0 & 0 & 0 & 0 \\ 0 & 0 & 0 & 0 & 0 \\ 0 & 0 & 0 & 0 & 0 \\ 0 & 0 & 0 & 0 & 0 \end{pmatrix}, \quad (4)$$

where $\frac{d\rho_0}{dt} = -W(t)\rho_0$ with $\rho_0$ being the population of Kr atom and $W(t)$ being the transient ionization rate. For complete incoherent injection, all the non-diagonal terms vanish. The term $[\rho^+]_{decay}$ contains the decay of population and coherence involved of the upper two states with Auger decay lifetime of 7.5 fs[35]. Once Eq. (1) is solved, the induced dipole moment by both the NIR and IAP can be obtained by

$$d(t,t_d) = \text{Tr}\left[\rho(t,t_d)\mu\right]. \quad (5)$$

The light absorption cross-section are calculated as[20]:

$$\sigma(\omega,t_d) = \frac{\omega}{c\varepsilon_0}\text{Im}\left[\frac{(d(\omega,t_d))}{E(\omega)}\right] = 4\pi\alpha\omega\text{Im}\left[\frac{(d(\omega,t_d))}{E(\omega)}\right] \quad (6)$$

where $c$ is the speed of light, $\varepsilon_0$ is the permittivity of vacuum and $\alpha$ is the fine-struture constant. $E(\omega)$ is the Fourier transform of the light pulse and $d(\omega,t_d)$ is the Fourier transform of $d(t,t_d)$.

## Data availability

The experimental and numerical data generated in this study have been deposited in the Zenodo database with access link of https://zenodo.org/records/10558809.

## Code availability

The simulation codes that support the findings of the study are available from the corresponding author upon request.

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

## Acknowledgements

This work was supported by the National Key Research and Development Program of China (Grant No. 2019YFA0307703), the NSF of China (Grant Nos. 12234020, 12274461, 11904400, 11974426), the Major Research Plan of the National Natural Science Foundation of China (91850201), and the Science and Technology Innovation Program of Hunan Province (No. 2022RC1193).

## Author contributions

X.W., J.Z., and Z.Z. discussed and conceived the idea. G.B., Q.Q., and J.Z. performed the simulations. L.W., J.W., F. X., W.T., P.S., and X.W. performed the experiments. J.L., J.Z., C.G. and Z.Z. analyzed the data. X.W., J.Z., and Z.Z. prepared the manuscript and discussed with all authors. L.W. and G.B. contribute equally to this work.

## Competing interests

The authors declare no competing interests.
