## [Peer Review File · Nature Communications]

REVIEWER COMMENTS

Reviewer #1 (Remarks to the Author):

Manuscript Wang et al reports on the population transfer and phase delay in absorption of two spin-orbit split ($4p\ 1/2^{-1}$ and $4p\ 3/2^{-1}$) ionic states in Kr. The effect is explained by the coherent Raman coupling to the $4s^{-1}$ state. Experimental measurement of the transient absorption is supported by detailed TDSE-calculations. The presented results of both experiment and theory are conclusive. The manuscript is of high relevance for the attosecond science / photon science community. It represents novel and essential contribution to the further development of the field. The manuscript is well written and meets expected standards. It strongly deserves publication in Nature Communication.

I have only one essential comment which should bring more transparency into the acquisition and analysis of experimental results. In case there is not enough space in the main manuscript authors should put this information in supplementary section. Authors should provide detailed information on calibration procedure of the spectrometer, delay-scan measurement – time-delay steps, averaging, reproducibility, background subtraction. Additionally, authors should consider to show raw data e.g. in supplementary and to include representative OD traces in Fig. 2

Reviewer #2 (Remarks to the Author):

Title: Raman time-delay in attosecond transient absorption of strong-field created Krypton vacancy

The authors have carried out a combined experimental and theoretical investigation of the response of Kr atoms to a few cycle infra-red laser pulse and an isolated attosecond pulse with a centre photon energy around 80 eV. Transient absorption is employed in this work, making use of a grating and a detector, which I presume is a toroidal grating and a microchannel plate detector.

The core observable is the change in optical density or absorbance which appear to be used interchangeably, as a function of the delay between the infra-red and XUV pulse. This quantity (absorbance w.r.t. time delay) is presented at two different intensities of the infra-red pulse, and a relative shift of time dependent structure is claimed to be the consequence of a Raman process between two light-coupled neighbouring states in the Kr^{2+} ion(?) My uncertainty here is a consequence of the labelling and layout of figure 1(b), which indicates the multiphoton coupling of the ground state of the Kr^{+} ion and the $4s_{-}(1/2)^{-1}$ at 27.5 eV above the ground state of the Kr neutral.

If this is observed as claimed, such a process will be of significance to the field of ultrafast spectroscopy. However, I feel there is a significant flaw in the data analysis which brings into question how “the full simulation essentially reproduces the experimental results...” (line 142). Looking at figure S4 in the supplementary material, two spectra are presented between 60 and 120 eV, comparing incident (blue/black) and transmitted (red) XUV pulses. No comment is made about the typical stability of the incident spectrum, and more importantly, the authors only concentrate on the spectral region between 78 to 82 eV (as seen in figure 2 in main manuscript) where there is a decrease in intensity between the incident and transmitted pulse, and no discussion is made of the

region 83 to 93 eV where there is an increase in intensity. Furthermore, the features identified as transitions to the excited states of the Kr⁺ ion are not ascribed error bars of any type, so while in figure 2 there is some indication that these are repeatable, the position (in energy), relative shape (in energy and time) and relative absorbance are, to my mind, not well quantified.

A similar issue is identified with Figure 3, whereby a comparison is made between two experimental absorbance lineouts corresponding to $4p_{3/2}^{-1}$ to $3d_{5/2}^{-1}$ and $4p_{1/2}^{-1}$ to $3d_{3/2}^{-1}$ at 79.8 and 80.4 eV and the output of theoretical considerations. As the experimental data is presented without reasonable error bars, it is impossible to assess how well the theoretical curves agree with the data or otherwise. The text claims a strong agreement, however I do not agree that such quantifications of relative delay, particularly on the attosecond timescale. This is compounded by no clear description being given of how the intensity of the infra-red radiation is found, particularly to three significant figures. In my experience, quantifying intensity to better than one significant figure is challenging, and this will directly impact the quality of the fit of the theoretical curves as it becomes essentially a free parameter.

Finally, in terms of reproducibility and experimental description, there is insufficient detail to allow another research group to repeat this experiment. Specifically, stability of pointing, intensity, spectrum of the IR source and corresponding XUV pulse, data collection times and error bars. The theoretical work associated with this work appears to be sound, however a fair and meaningful conclusion cannot be reached without a more complete description and quantification of the presented experimental observations.

Reviewer #3 (Remarks to the Author):

This manuscript reports the use of attosecond transient absorption spectroscopy to investigate the strong-field ionization of krypton. The experimental data reveals time delay shifts of a few hundred attoseconds between the resonant absorption signals of the $4p_{3/2-1}$ and $4p_{1/2-1}$ spin-orbit states of the Kr⁺ ion. Accompanying theoretical simulations suggest that the time delay shift originates from the coherent coupling of states. While the experimental data is of high quality and the interpretation of the data is supported by theoretical simulations, I am concerned about the lack of novelty in the work. Specifically, the work appears to be very similar to that reported Nat. Phys. 13, 472–478 (2017), cited as ref. 25 in the manuscript, the only difference being Kr being used in the present work and Xe being used in the previous work. In ref. 25, an intensity-dependent time delay in the resonant absorption signals of the $5p_{3/2-1}$ and $5p_{1/2-1}$ spin-orbit states of the Xe⁺ ion was reported and explained in terms of the laser-induced electronic polarization of the Xe atom. Compared to ref. 25, it is unclear what new insight the current work provides. Aside from the lack of novelty, the simulation of the transient absorption signal also does not consider the temporal nonlocal nature of absorption spectroscopy, discussed in Phys. Rev. A 83, 033405 (2011), cited as ref. 20 of the manuscript. This nonlocal nature needs to be considered for accurate simulation of the transient absorption signal in the region of pump-probe temporal overlap. For these reasons, I cannot recommend publication of this manuscript in Nature Communications.

RESPONSES TO REVIEWERS

We sincerely appreciate all three reviewers for their valuable time and effort to the evaluation of our work. We have thoroughly revised the manuscript according to their constructive comments, in addition to new analyses to reinforce the robustness of our findings. Our detailed responses to each of the reviewers' comments can be found in **red**, and the revisions made to our manuscript are shown in **blue**.

Reviewer #1:

Manuscript Wang et al reports on the population transfer and phase delay in absorption of two spin-orbit split ($4p\ 1/2^{-1}$ and $4p\ 3/2^{-1}$) ionic states in Kr. The effect is explained by the coherent Raman coupling to the $4s^{-1}$ state. Experimental measurement of the transient absorption is supported by detailed TDSE-calculations. The presented results of both experiment and theory are conclusive. The manuscript is of high relevance for the attosecond science / photon science community. It represents novel and essential contribution to the further development of the field. The manuscript is well written and meets expected standards. It strongly deserves publication in Nature Communication.

We thank the reviewer for his/her supportive comments highlighting the significance of our work and recommending the publication of the manuscript.

Comment:

I have only one essential comment which should bring more transparency into the acquisition and analysis of experimental results. In case there is not enough space in the main manuscript authors should put this information in supplementary section. Authors should provide detailed information on calibration procedure of the spectrometer, delay-scan measurement – time-delay steps, averaging, reproducibility, background subtraction. Additionally, authors should consider to show raw data e.g. in supplementary and to include representative OD traces in Fig. 2.

Reply:

We thank the reviewer's helpful comments. The details of data acquisition and analysis are important to ensure the creditability of our data and arguments. As space was restricted in the main manuscript, we have added the specifics of the spectrometer calibration, the delay-scan measurement, the reproducibility, the background subtraction, and the raw data of Fig.2 in the revised supplementary materials.

1. Calibration of Spectrometer:

The XUV spectrometer we used in the experiments consisted of a flat-field grating (Hitachi 001-0660) and a microchannel plate detector (Tectra), as shown in Fig. R1(a). The transmitted XUV beam after the gas cell illuminated the grating with a grazing angle of $\alpha = 87^\circ$, and the diffraction angle β satisfies:

$$\sin\alpha - \sin\beta = m\lambda G, \quad (\text{R1})$$

where $m = 1$, λ , and $G = 1200 \text{ mm}^{-1}$ are the diffraction order, wavelength and grating constant, respectively. Due to specially designed variable groove space and surface curvature, the spectral components with wavelength between 5-25 nm are focused on a flat focal plane located $L = 563.2 \text{ mm}$ away from the grating. The wavelength of the diffracted spectral component located at position y on the focal plane can be written as:

$$\lambda = \frac{1}{mG} \left(\sin\alpha - \sqrt{\frac{L^2}{y^2 + L^2}} \right). \quad (\text{R2})$$

It suggests a very complex dependence of wavelength λ on position y . However, it can be approximated by a 3rd order polynomial, as shown in Fig. R1(b). Therefore, the spectrometer was calibrated by fitting some known spectral lines with a 3rd order polynomial.

Fig. R1 Calibration of the XUV spectrometer. (a) The XUV light from the source, i.e., the absorption cell in our experiments, is diffracted by a flat-field grating (Hitach 001-0660) with incident angle of $\alpha = 87^\circ$, forming a flat focal plane located 563.2 mm away from the grating. The MCP detector is placed right on the focal plane to record the spectrum. The Zr foil, which is

used to remove NIR laser pulses, is mounted on the center of a vacuum valve installed on the entrance of the spectrometer. (b) The dependence of diffracted spectral component wavelength on position y on the focal plane can be described by a 3rd order polynomial. (c) The spectrometer was calibrated with three transition lines located at 79.8 eV, 80.4 eV and 81.1 eV (circles), together with discrete harmonic lines (squares).

The three transitions $4p_{3/2}^{-1} \rightarrow 3d_{5/2}^{-1}$, $4p_{1/2}^{-1} \rightarrow 3d_{3/2}^{-1}$ and $4p_{3/2}^{-1} \rightarrow 3d_{3/2}^{-1}$ located at 79.8 eV, 80.4 eV and 81.1 eV respectively were observable on the spectrometer, as shown by the filled circles in Fig. R1(c). Besides, discrete harmonics (squares in Fig. R1(c)) generated with linear polarized long pulses were used for calibration as well. All the spectral lines were then fitted with a 3rd order polynomial as explained before. Note that the fitting was done to convert pixel to wavelength, then photon energy was obtained from the wavelength.

2. Delay-scan measurement:

Fig R2: Active delay stabilization. (a) Interference fringes recorded over 856 seconds when the pump-probe delay is stabilized and scanned. (b) The phase of the fringes extracted from (a) for each delay time. The step-like phase change reflects the pump-probe delay adjustment with

scanning step of 0.5 rad (141 as). (c) The relative phase, i.e., the error between actual phase and target phase, indicates a rms error of 0.08 rad (23.8 as).

In regard to the delay scan measurements, which is the key to ensure the accurate temporal resolution, we used active delay control technique. A continuous green beam was split by the same beam splitter in ATAS setup, so that the two green beams went through the XUV and NIR path, and recombined on the hole-mirror to form interference fringes. By stabilizing the fringes with PID feedback electronics, the pump-probe delay can be stabilized. In Fig. R2(a), we show an example delay scanning fringes which last for 856 seconds. The pump-probe delay can be obtained from the phase of the fringes via $\tau = \frac{\phi T_0}{2\pi}$, where $T_0 = 1.77$ fs is the period of 532 nm green light used in the delay locking setup. The phase of the interference fringes is shown in Fig. R2(b). It is shown that the phase changed 0.5 rad per scan step, i.e. the delay scan step was 141 as. To evaluate the time jitter, we should pay attention to the relative phase change, i.e. the error between the actual phase and target phase, as shown in Fig. R2(c). The rms phase error was 0.08 rad, which suggested a time jitter of 23.8 as. During each scanning step, the transmitted spectra were integrated for 20 seconds, which means 20000 spectra were averaged.

However, the actual time resolution of the electron dynamics extracted from ATAS spectra can be much higher than either the scan step or delay jitter, since the sinusoidal evolution can be rebuilt with sparse sampling points. For example, 5 as precision was achieved with <30 as jitter and 170 as step size in ATAS measurements previously [Opt. Lett., 44, 4749, 2019]. By adopting the similar Monte Carlo simulations [Philos. Trans. Royal Soc. A, 377, 20170475, 2019] for RABITT measurements, the temporal resolution of ATAS can also be analyzed. In the simulations as shown in Fig. R3(a), we sampled a sinusoidal curve (blue line) with 141 as scan step for different time jitter and scan length, then the sampled points (red circles) were fitted by a sinusoidal curve (red line). The temporal error was obtained by comparing the fitted curve with the true curve. The temporal resolutions, i.e., the statistic error over 20000 simulations, for different time jitter and different scan length are shown in Fig. R3(b). It is shown the precision becomes better as time jitter decreases and scan length increases. Especially, for 25 as time jitter and 2 optical cycle scan length as were the case in our experiments, the temporal resolution can be as high as 8 as.

Fig. R3: Temporal resolution analysis for ATAS measurements. (a) a sinusoidal curve (blue line) was sampled with 141 as scan step for a certain time jitter and scan length. The sampled points (red circles) were fitted by a sinusoidal curve (red line). The phase (temporal) difference between the fitted curve and the actual curve gives the temporal precision. (b) The dependence of the precision on time jitter and scan length. Especially, for 25 as time jitter and 2 optical cycles of fundamental pulses, the temporal resolution can be as high as 8 as.

3. Reproducibility and background subtraction

Long integration time (20 s) were applied during the spectrum recording for each delay to ensure the signal-to-noise level and data fidelity. In addition, the experiments were carried out for multiple times, and the observed dynamics were reproducible. Fig. R4 shows four additional datasets of ΔOD , in which the oscillations in the absorption of two transition channels and their relative delay were evident.

Fig. R4: Four additional datasets for ATAS measurements. The oscillations in the absorption of two transition channels and their relative delay were evident, showing good data reproducibility.

Due to laser pointing instability, power fluctuation, thermal expansion of mirror mounts, air turbulence, and any other random changes of the system, the acquired transmission spectra over long time are always noisy, hence, background subtraction is usually needed. The random spectra change occurs in every scan step, manifesting as high frequency oscillations in the ATAS spectrum. Therefore, it is reasonable to suppress the fast oscillational noise by smoothing the raw data. In the smoothing procedure, each data point is updated by the average of its neighboring data points within a moving window in which Gaussian weights are applied to the data points. As shown in Fig. R5, the raw data (dots) can be improved significantly in terms of oscillational noise removing by performing Gaussian weighted smoothing (solid curve).

Fig. R5: Data smoothing with Gaussian-weighted moving average method. The experimental data (dots) were smoothed by averaging its neighboring points within a Gaussian window. The smoothed data (solid line) show less structures comparing to the raw data.

4. Raw data

The raw data of the two presented datasets in the main manuscript are shown in Fig. R6(a) and (b) for IR intensity of $3.6 \times 10^{14} \text{ W/cm}^2$ and $4.6 \times 10^{14} \text{ W/cm}^2$, respectively. By performing the Gaussian-weighted averaging for delay-dependent ΔOD in both the delay axis (3 data points window width) and energy axis (2 data points window width), we obtained the background-free traces, as shown in Fig. R6(c) and (d), respectively. These are the data sets demonstrated in the main manuscript.

Fig. R6: The raw data (a,b) and smoothed background-free data (c,d) of the ATAS traces in the main text for IR intensity of $3.6 \times 10^{14} \text{ W/cm}^2$ (a,c) and $4.6 \times 10^{14} \text{ W/cm}^2$ (b,d). The smoothing was done with Gaussian-weighted moving average method with window with of 2 data points and 3 data points for the energy axis and delay axis, respectively.

The changes of OD traces for the transitions of $4p_{3/2}^{-1} \rightarrow 3d_{5/2}^{-1}$ and $4p_{1/2}^{-1} \rightarrow 3d_{3/2}^{-1}$ were shown in Fig. 3 of the main manuscript. And they were compared with the theoretical calculations. So we prefer not to include any ΔOD traces in Fig. 2. Instead, in the revised manuscript, we updated Fig. 2 with clearer colormap and added labels for the three transition channels.

Reviewer #2:

The authors have carried out a combined experimental and theoretical investigation of the response of Kr atoms to a few cycle infra-red laser pulse and an isolated attosecond pulse with a centre photon energy around 80 eV.

Comment No.1:

Transient absorption is employed in this work, making use of a grating and a detector, which I presume is a toroidal grating and a microchannel plate detector.

Reply:

We thank the reviewer for his/her comments. The XUV spectrometer consists of a spherical variable line space grating (Hitachi, 001-0660) and a microchannel plate detector (Tectra). The details of the spectrometer and its calibration, which were explained previously in this reply letter, were included in the revised supplementary material.

Comment No.2:

The core observable is the change in optical density or absorbance which appear to be used interchangeably, as a function of the delay between the infra-red and XUV pulse. This quantity (absorbance w.r.t. time delay) is presented at two different intensities of the infra-red pulse, and a relative shift of time dependent structure is claimed to be the consequence of a Raman process between two light-coupled neighbouring states in the Kr^{2+} ion(?) My uncertainty here is a consequence of the labelling and layout of figure 1(b), which indicates the multiphoton coupling of the ground state of the Kr^+ ion and the $4s_{1/2}^{-1}$ at 27.5 eV above the ground state of the Kr neutral.

Reply:

We thank the reviewer for pointing out the ambiguous statements in the manuscript. In the revised manuscript, we used “change of optical density” throughout the paper. There are three kinds of spectra measured in the experiments: 1) the spectrum for the input IAP with spectral intensity of S_x^0 , 2) the transmitted spectrum after neutral Kr atoms S_x^n and 3) the transmitted spectrum of Kr^+ ion S_x^i . The change of optical density ΔOD is defined as:

$$\Delta OD = -\log \frac{S_x^i}{S_x^0} + \log \frac{S_x^n}{S_x^0} = -\log \frac{S_x^i}{S_x^n} \quad (\text{R3})$$

Therefore, the uncertainty of ΔOD can be evaluated as:

$$e_{\Delta OD} = \sqrt{\left(\frac{\partial \Delta OD}{\partial S_x^i} dS_x^i\right)^2 + \left(\frac{\partial \Delta OD}{\partial S_x^n} dS_x^n\right)^2} = \sqrt{\left(\frac{dS_x^i}{S_x^i}\right)^2 + \left(\frac{dS_x^n}{S_x^n}\right)^2} \quad (\text{R4})$$

where dS_x^i and dS_x^n is the uncertainty of measured transmitted spectra after Kr^+ ion and neutral Kr atoms, respectively.

The Raman process we discussed in the manuscript involved two light-coupled neighboring states of Kr^+ , and Kr^{2+} ions were not involved. We have changed the relevant statements in the manuscript to make this clearer. Figure 1(b) was updated as well accordingly in the revised manuscript. And it is true that we demonstrated the multiphoton coupling between the ground state and excited state ($4s_{1/2}^{-1}$) of Kr^+ ion, and the energy spacing between them is as high as ~ 13 eV.

Comment No.3:

If this is observed as claimed, such a process will be of significance to the field of ultrafast spectroscopy. However, I feel there is a significant flaw in the data analysis which brings into question how “the full simulation essentially reproduces the experimental results...” (line 142). Looking at figure S4 in the supplementary material, two spectra are presented between 60 and 120 eV, comparing incident (blue/black) and transmitted (red) XUV pulses. No comment is made about the typical stability of the incident spectrum, and more importantly, the authors only concentrate on the spectral region between 78 to 82 eV (as seen in figure 2 in main manuscript) where there is a decrease in intensity between the incident and transmitted pulse, and no discussion is made of the region 83 to

93 eV where there is an increase in intensity. Furthermore, the features identified as transitions to the excited states of the Kr⁺ ion are not ascribed error bars of any type, so while in figure 2 there is some indication that these are repeatable, the position (in energy), relative shape (in energy and time) and relative absorbance are, to my mind, not well quantified.

Reply:

We thank the reviewer for pointing out the flaw of our data analysis and demonstration. The statement “the full simulation essentially reproduces the experimental results...” was made according to Fig. 3 of the main text, and we agree with the reviewer that without error bars for the experimental data, it is hard to conclude a good consistency. According to Eq. (R4), the error bars of the ΔOD was calculated, as shown in Fig. R7. In addition to the error bars, the experimental data (scatters) are also fitted with a smooth sinusoidal curve $f(\tau) = g(\tau) \cos(2\pi f_0 \tau + \phi) + h(\tau)$, as shown by the thinner solid lines. Here, $g(\tau)$ and $h(\tau)$ are two 6-order polynomials to adjust the amplitudes and positions of the sinusoidal oscillation, f_0 is oscillation frequency and ϕ is the phase of the curve. As demonstrated in Fig. R3, the fitting procedure allows to improve the time resolution to as high as 8 as for the delay control parameters used in our experiments. The delay between different transition paths is then determined to be 420 as and 480 as for low ($3.6 \times 10^{14} \text{ W/cm}^2$) and high ($4.6 \times 10^{14} \text{ W/cm}^2$) laser intensity, respectively. Previously, the delays (423 as and 564 as) were obtained from the discrete experimental data points, so they can only be integral multiples of delay scan step (141 as). In the revised figure, the delays are more reasonable and accurate.

In Fig. R7(a), the simulated ΔOD for the transition of $4p_{3/2}^{-1} \rightarrow 3d_{5/2}^{-1}$ (red thicker solid line) and $4p_{1/2}^{-1} \rightarrow 3d_{3/2}^{-1}$ (green thicker solid line) with laser intensity of $3.6 \times 10^{14} \text{ W/cm}^2$ is compared with the corresponding experimental data (scatters). For higher laser intensity of $4.6 \times 10^{14} \text{ W/cm}^2$, the same comparison is shown in Fig. R7(b). It is shown that the simulations reproduce the experiments qualitatively. Although the absolute ΔOD was not accurately reproduced, the oscillation features and the relative delay between the two transitions are predicted satisfactorily by the theory.

R7: Revised figure 3 for the main manuscript. The experimental ΔOD (scatters) for NIR laser intensity of $3.6 \times 10^{14} \text{ W/cm}^2$ (a) and $4.6 \times 10^{14} \text{ W/cm}^2$ (b). The experimental data points were fitted with smooth sinusoidal curve (thinner solid lines) to find the relative delay between the transitions $4p_{3/2}^{-1} \rightarrow 3d_{5/2}^{-1}$ (red) and $4p_{1/2}^{-1} \rightarrow 3d_{3/2}^{-1}$ (green). And theoretical calculations (solid curves) reproduce the ΔOD qualitatively for both the laser intensities and both the transitions.

In Fig. S4 of the supplementary material, the two spectral curves were normalized to their maximum value respectively, i.e. the two curves were scaled with two different factors. Therefore, the direct spectral intensity comparison between them was misleading. We reprocessed the spectra and revised the figure with unnormalized data, as shown in Fig. R8. The red curve represents the spectrum of XUV pulses. The blue curve shows the spectrum of transmitted XUV pulse through neutral Kr atoms, and the absorption spectrum of Kr^+ is shown as the green curve. In the region 83 to 93 eV, the spectral intensity decreases as well when Kr^+ ions are produced. The stability of the three spectra were evaluated from multiple measurements and plotted as filled area around the curves. The uncertainty of transmitted spectra after Kr^+ ion and neutral Kr atoms dS_x^i and dS_x^n can thus be obtained, and the uncertainty of ΔOD can be evaluated from Eq. (R4).

Fig. R8: The spectra of XUV pulses for the input IAP pulses (red) and the transmitted pulses through Kr (blue) and Kr⁺ (green).

Comment No.4:

A similar issue is identified with Figure 3, whereby a comparison is made between two experimental absorbance lineouts corresponding to $4p_{(3/2)^{-1}}$ to $3d_{(5/2)^{-1}}$ and $4p_{(1/2)^{-1}}$ to $3d_{(3/2)^{-1}}$ at 79.8 and 80.4 eV and the output of theoretical considerations. As the experimental data is presented without reasonable error bars, it is impossible to assess how well the theoretical curves agree with the data or otherwise. The text claims a strong agreement, however I do not agree that such quantifications of relative delay, particularly on the attosecond timescale. This is compacted by no clear description being given of how the intensity of the infra-red radiation is found, particularly to three significant figures. In my experience, quantifying intensity to better than one significant figure is challenging, and this will directly impact the quality of the fit of the theoretical curves as it becomes essentially a free parameter.

Reply:

We thank the reviewer for the comments. Figure 3 was updated with error bars for the experimental data, as shown in Fig. R7. And the agreement between the experimental and theoretical results become more intuitive and convincing. The intensity value with three significant figures in the main text is non-professional, and was corrected in the revised version. The NIR laser intensity was calibrated with the cut-off energy of high harmonics generated by Kr atoms. As show in Fig. R9, the cut-off energy of I_1 and I_2 is about $71.8(\pm 3.4)$ eV and $87.1(\pm 3.4)$ eV, respectively. The uncertainty of the cut-off energy comes from the ambiguous cut-off position determination. According to the relationship between high harmonic cut-off energy E_c and driving laser intensity I , the laser intensity can be obtained with:

$$I \approx \frac{E_c - I_p}{3.17 \times 9.33 \lambda^2}$$

Where I_p (eV) is the ionization potential of Kr atoms, λ (μm) is the wavelength of the driving laser.

The laser central wavelength is 733 ± 20 nm, then the estimated laser intensity for I_1 and I_2 is $3.6(\pm 0.3) \times 10^{14} \text{W/cm}^2$ and $4.6(\pm 0.3) \times 10^{14} \text{W/cm}^2$, respectively.

Fig. R9: Determination of laser intensity of NIR pulses with the cut-off energy of HHG.

Comment No.5:

Finally, in terms of reproducibility and experimental description, there is insufficient detail to allow another research group to repeat this experiment. Specifically, stability of pointing, intensity, spectrum of the IR source and corresponding XUV pulse, data collection times and error bars. The theoretical work associated with this work appears to be sound, however a fair and meaningful conclusion cannot be reached without a more complete description and quantification of the presented experimental observations.

Reply:

We thank the reviewer for the comments. The experimental details were explained in detail in the revised manuscript/supplementary material, as shown in Fig. R10. The laser for the experiments was a 1 kHz multi-pass chirped pulse amplification system (Femtopower HE), which delivers 25 fs, 4.2 mJ pulses. The multi-cycle pulses were then coupled to a 1-m long helium-filled hollow-core fiber (HF) with diameter of $300 \mu\text{m}$. To decrease the ionization effect and improve the throughput of the fiber, differential pumping scheme was employed. High pressure helium gas (2000 mbar) was introduced from the exit of the fiber, while pressure of the entrance was kept as low as $3\text{E-}2$ mbar with a dry pump (TriScroll 300). The spectrum was then broadened to cover a spectral range of 580-940 nm, as shown in Fig R11(b). To get ultrashort few-cycle pulses, chirped mirrors (Laser Quantum DCM7) were used for group delay dispersion compensation. After the chirped mirrors, the NIR pulse

was then compressed down to 5.3 fs, which was measured with a fringe resolved autocorrelator (Femtometer), as shown in Fig. R11(a). The FWHM of the autocorrelation signal is 10 fs, so the duration of NIR pulses is estimated to be 5.3 fs assuming sech² temporal shape.

Fig. R10: Detailed experimental layout. HF: 1-m long hollow-core fiber; BS: 50:50 ultrafast beam splitter; QP1: 177 μm thick quartz plate; QP2: 440 μm thick quartz plate; BBO: 141 μm thick BBO crystal; FM: silver-coated focal mirror; GC1: the first gas cell for IAP generation; TM: toroidal mirror; HM: hole mirror; SM: flat silver mirror; FL: focal lens, GC2: the 2nd gas cell for absorption measurements.

The few-cycle pulses after the chirp mirrors with pulse energy of 1.8 mJ were divided equally into two parts by a 50% beam splitter (FemtoOptics OA037). The electric field of one arm was manipulated to form a half laser cycle gate for IAP generation with DOG optics, which consisted of a 177 μm thick quartz plate (QP1), a 440 μm thick quartz plate (QP2) and a 141 μm thick beta-barium borate (BBO) crystal. The pulses were then focused into neon gas filled cell by a concave mirror with focal length of 350 mm to generate IAP. A 200 nm thickness Zr foil (Lebow) was used to block residual NIR beam. The IAP was then focused by a gold coated toroidal mirror (CLaser) into the second gas cell filled with Kr samples (50 mbar). The other arm from the beam splitter was propagated along a delay line installed on a nano-precision piezo-stage (Physik Instrumente, P-752.1CD) and combined with IAP by a hole-drilled mirror (HM). The pump beam was focused by a lens with focal length of 400 mm to generate vacancy in Kr valance shell. The IAP serves as the probe of the vacancy generation dynamics. After interacting with Kr gas, the residual NIR was filtered out again using a 200 nm thickness Zr foil (Lebow), and the transmitted IAP was recorded by a home-

made XUV spectrometer. The time delay between the NIR and the IAP was stabilized and controlled with active phase locking technology, to implement which a green continuous-wave (cw) laser beam was sent through the two arms of the pump-probe system and formed interference fringes after the HM mirror. The piezo-stage was adjusted by a proportional-integral-differential (PID) control procedure according to the movement of the interference fringes. The stability of the pump-probe delay was 23.8 as, and the scan step size was 141 as, as shown in Fig. R2.

Fig. R11: (a) The fringe resolved autocorrelation signal of the compressed few-cycle pulses after chirped mirrors. The FWHM of the autocorrelation signal is 10 fs, so the duration of NIR pulses is estimated to be 5.3 fs assuming sech2 temporal shape. (b) the spectrum of the broadened pulses after hollow-core fiber.

The pointing stability of the NIR laser pulses was measured by monitoring the focal spot position with a charge-coupled device (CCD) over 120 minutes, as shown in Fig. R12(a). The pointing stability was better than $20 \mu\text{rad}$. The power stability after hollow-core fiber was measured with a power-meter (Thorlabs PM100D), as shown in Fig. R12(b). The rms error of the pulse energy is estimated to be 1.1%.

Fig. R12: The NIR laser pointing stability (a) and power stability. The pointing stability was better than 20 μrad , while the rms error of the pulse energy is estimated to be 1.1%.

The spectrum of the XUV IAP pulses and IAP energy stability are shown in Fig. R8 (red curve). Since the wavefront of the IAP pulses is determined by that of the NIR pulses, the IAPs are expected to have the same pointing stability as the NIR pulses.

Reviewer #3:

This manuscript reports the use of attosecond transient absorption spectroscopy to investigate the strong-field ionization of krypton. The experimental data reveals time delay shifts of a few hundred attoseconds between the resonant absorption signals of the $4p_{3/2-1}$ and $4p_{1/2-1}$ spin-orbit states of the Kr^+ ion. Accompanying theoretical simulations suggest that the time delay shift originates from the coherent coupling of states. While the experimental data is of high quality and the interpretation of the data is supported by theoretical simulations.

We thank the reviewer for his/her positive remarks on our experimental and theoretical results.

Comment No.1:

I am concerned about the lack of novelty in the work. Specifically, the work appears to be very similar to that reported Nat. Phys. 13, 472–478 (2017), cited as ref. 25 in the manuscript, the only difference being Kr being used in the present work and Xe being used in the previous work. In ref. 25, an intensity-dependent time delay in the resonant absorption signals of the $5p_{3/2-1}$ and $5p_{1/2-1}$ spin-orbit states of the Xe^+ ion was reported and explained in terms of the laser-induced electronic

polarization of the Xe atom. Compared to ref. 25, it is unclear what new insight the current work provides.

Reply:

We thank the reviewer for the comments. We believe that our work represents a significant advancement and provides valuable insights into understanding laser-dressed bound electron motion on the fundamental timescale, thereby greatly enhancing attosecond chronoscopy capabilities, as judged by Referee 1 that “it represents novel and essential contribution to the further development of the field”. In the following, we would like to summarize the essential differences of our work from Ref. 25.

Although Ref. 25 and our present work both involve attosecond transient absorption spectroscopy and exhibit certain similarities in the experimental observations, they actually investigate distinct processes and the underlying physics. In ref. 25, the authors attributed the half-cycle modulation of the absorption signal to the transient ground-state polarization of the neutral Xe. As stated in ref. 25 *“Although this study provides an intuitive understanding of sub-cycle features in strong-field ionization, further investigations need to be taken to shed light onto the non-trivial origin of the observed phase delay”*, our manuscript focuses right on the understanding the phase delay in the absorption spectra between the two spin-orbit split states. By considering the coupling of the ions with the remaining laser field, we demonstrate that the observed time delay results from the Raman process between the ionic state $4s^{-1}$ and the two spin-orbit split states $4p_{3/2}^{-1}$ and $4p_{1/2}^{-1}$. Moreover, in comparison to the experiments conducted with Xe (where $\alpha_0 \approx 27$ a.u.) in ref. 25, the Kr atoms are more challenging to probe technically due to their significantly lower polarizability ($\alpha_0 \approx 17$ a.u.).

To emphasize the distinctiveness of our work, we highlight the following key points:

1. Introduction of the Raman time delay: We propose and experimentally demonstrate the concept of the Raman time delay, which is fundamentally different from the results in ref. 25. We experimentally observe that the absorptions by the two spin-orbit split states are modulated at different paces when varying the time-delay between the infrared pumping pulse and the attosecond probing pulse. Through detailed theoretical analysis, we find that the two spin-orbit split states are coupled to ionic $4s^{-1}$ state with multiphoton resonant transitions induced by the remaining pump pulse. This Raman process explains the non-trivial origin of the time delay.

2. On the transient ground-state polarizability: In ref. 25, the half-cycle modulation of the absorbance is attributed to the neutral ground-state polarizability, which we also incorporate in the theoretical simulation. As mentioned in line 144 of the manuscript, “the reversible ionization induced by the polarization of the neutral ground state in ref.25 is reproduced by our simulation, with the ionization probability calculated by the strong field approximation (SFA) instead of the ADK model.” Since the SFA theory includes the quantum coherence between the continuum states and the ground

state, the polarization of the neutral atom is naturally included in the time-evolving wave function whereas the coupling with the neutral excited states has been neglected.

3. Role of quantum coherence: Our investigation underscores the critical role played by quantum coherence in analyzing transient absorption spectra of the ions induced by strong-field ionization. The transient ionization injection creates coherence among different ionic states, leading to population transfer between the two spin-orbit split states with their relative rate agreeing better with the experimental measurements. Moreover, we demonstrate that ions are produced with less coherence at higher intensities and longer pulse lasers, owing to the lack of phase synchronization between the coherence injected via ionization at different optical cycles or transient ionization.

Comment No.2:

Aside from the lack of novelty, the simulation of the transient absorption signal also does not consider the temporal nonlocal nature of absorption spectroscopy, discussed in Phys. Rev. A 83, 033405 (2011), cited as ref. 20 of the manuscript. This nonlocal nature needs to be considered for accurate simulation of the transient absorption signal in the region of pump-probe temporal overlap.

Reply:

We thank the reviewer for pointing out this critical question in the absorption spectra. Indeed, the transient absorption inherently measures the response during the whole interaction time, rather than the instantaneous effect, which is exactly included in our simulation. We calculate the absorption spectrum from the time-dependent dipole moment within the entire interval during interaction with the infrared laser pulse and the XUV pulse (see Eq. (2) in the manuscript), indicating that the time delays in the resonant absorptions are inherently non-local in time. This can also be seen clearly in Fig. 4 in the manuscript, where the time delays of the instantaneous population modulations between the two spin-orbit split states are only 45 and 75 attoseconds at two laser intensities. In contrast, the time delays retrieved from the experimentally measured transient absorption spectra are 200 and 250 attoseconds, which highlights the temporal nonlocal nature of the absorption spectra. In addition, we would comment that the non-local absorption and re-emission would lead to the variation of absorption profile as evidenced by the evolution of the Fano profile observed experimentally by us and many other groups.

In summary, our work reports novel insight into the fascinating understanding of the laser-ion interactions enriching attosecond chronoscopy. We would highly appreciate your reconsideration and recommendation of our findings.

Main changes of the revised manuscript:

- 1. Figure 1 was updated to make the electron transition and states coupling clearer;**
- 2. Figure 2 was updated with energy labels to indicate the three transition channels;**
- 3. Figure 3 was updated to include error bars for experimental data, and the relative time delay between the two transitions was recalculated from the fitted curve rather than the raw experimental data;**
- 4. Figure 4 was updated to make the time delay differences clearer;**
- 5. All the changes regarding to the reviewers' comments/questions are marked in red;**
- 6. All the other changes including typos, gramma error correction, language polishing, and additional details are marked in blue.**

REVIEWER COMMENTS

Reviewer #1 (Remarks to the Author):

Authors have taken seriously my criticism and have revised their manuscript accordingly. I do not have any further comments. The manuscript can now be accepted.

Reviewer #2 (Remarks to the Author):

The authors present a modified manuscript describing their investigations into "Raman time-delay in attosecond transient absorption of strong-field created krypton vacancy", with the authors making solid efforts to address the comments from the three reviewers. This has made the manuscript far more straightforward to grasp, with the much improved manuscript, method and supplementary information providing illustration that the methodology is in line with standards of contemporary works.

I feel that the results will be of some significance to the field. With that said, there is overlap with previous studies, and unfortunately I am not completely convinced that their findings are noteworthy.

Reviewer #3 (Remarks to the Author):

The authors have provided a detailed response to the previous comments of the reviewers. In most cases, the responses appear to be satisfactory. However, I still have difficulty seeing the novelty of the current manuscript. Both ref. 25 and the current manuscript report (1) the appearance of half-cycle modulations in the transient absorption signal of ions (Xe⁺ in ref. 25 and Kr⁺ in this manuscript) produced by strong-field ionization and (2) intensity-dependent phase shifts between the half-cycle modulations observed at the absorption resonances of the two spin-orbit states. Hence, I disagree with the authors' claim that the two studies "investigate distinct processes and the underlying physics." Moreover, I note that ref. 25 reproduces the experimental observation of both (1) & (2) by using TDCIS simulations. While ref. 25 does not pinpoint the precise origin of the phase shift – they speculate that it arises from nonadiabaticity of the strong-field ionization – I am not entirely convinced that the present manuscript does. For one, the agreement between experiment and theory is not particularly satisfactory; it is evident from Figs. 3a and 3b that the theoretically predicted phase shift (~ 0.2 fs) is smaller than the experimentally observed phase shift ($\sim 0.4 - 0.5$ fs). Second, the authors only consider coupling to the $4s-1$ state, even though other ion states exist in its vicinity. Third, it is not clear if the simulations account for the nontrivial relation between population and absorption signal in the temporal region of pump-probe overlap; see Phys. Rev. A 86, 063411 (2012). Briefly, due to the finite lifetime of the Kr 3d core hole, the transient absorption probe also samples the time-dependent population after pump-probe time delay t_d . Hence, the transient absorption signal at pump-probe time delay t_d does not directly reflect the ion population

at td. For the above reasons, in addition to the lack of novelty, I recommend publication of the manuscript in a more specialized physics journal. There, the authors could elaborate on the differences between the behaviors of Kr and Xe, expand their discussion of laser-coupling involving the $4s-1$ states to include other ion excited states, address why they think that nonadiabatic effects do not explain the phase shift (as postulated by the authors in ref. 25), and propose other possible mechanisms that might give rise to the phase shift.

RESPONSES TO REVIEWERS

We sincerely appreciate all the three reviewers for their valuable time and effort spent on the re-evaluation of our work. Our detailed responses can be found in green, and the revisions made to our manuscript are shown in blue.

Reviewer #3:

Q1: The authors have provided a detailed response to the previous comments of the reviewers. In most cases, the responses appear to be satisfactory. However, I still have difficulty seeing the novelty of the current manuscript. Both ref. 25 and the current manuscript report (1) the appearance of half-cycle modulations in the transient absorption signal of ions (Xe⁺ in ref. 25 and Kr⁺ in this manuscript) produced by strong-field ionization and (2) intensity-dependent phase shifts between the half-cycle modulations observed at the absorption resonances of the two spin-orbit states. Hence, I disagree with the authors' claim that the two studies "investigate distinct processes and the underlying physics."

R1:

We regret that we did not successfully clarify the difference between our work and Ref. 25 in our previous response. Although the two works appear reporting two above-mentioned similar phenomena, **the novelty of this work lies in the effects of Raman coupling and the quantum coherence on the resonant absorptions**, which have been ignored in previous investigations but uncovered with our new theoretical approaches. As indicated by the title of our manuscript, this work focuses particularly on the attosecond chronoscope concept of the measuring the time-delays between the resonant absorptions of the nonstationary Kr⁺. In contrast, Ref. 25 discussed the time delay of the reconstructed density matrix rather than the absorption lines. The differences can be highlighted: (i) the physical origin of the phase shift was not identified in Ref. 25 due to the complexity and the less-transparency of TDCIS, which has other limitations (see R2 for details); (ii) we have pointed out three distinct processes that come into play, i.e., the coherence of ionization, the coherence-driven population transfer and the Raman time-delay. It should be pointed out that the theoretical explanation of time delays is challenging for the attosecond community, for example, the explanation on the tunneling time delays and the photoionization time delays are still highly debated. For the time delays measured in the complex ionic system created by strong-field ionization, a more transparent theory is essential to interpret the underlying physics. In all, we believe this work is concept-breaking and validated by solid theory and state-of-art attosecond transient absorption measurements.

Q2: Moreover, I note that ref. 25 reproduces the experimental observation of both (1) & (2) by using TDCIS simulations. While ref. 25 does not pinpoint the precise origin of the phase shift – they speculate that it arises from nonadiabaticity of the strong-field ionization – I am not entirely

convinced that the present manuscript does. For one, the agreement between experiment and theory is not particularly satisfactory; it is evident from Figs. 3a and 3b that the theoretically predicted phase shift (~ 0.2 fs) is smaller than the experimentally observed phase shift ($\sim 0.4 - 0.5$ fs). Second, the authors only consider coupling to the $4s-1$ state, even though other ion states exist in its vicinity.

R2:

Thank you for your valuable comments that would help us improve our work. Although TDCIS gives a general recipe for dealing with the dynamics, the reliability of the calculation depends on the accurate description of the configurations, the full continuum space and the unavoidable boundary absorption, which can be hardly met. In Ref. 25, the time-delay is found 75 attoseconds from the TDCIS calculation, while their experiment gives 200 attoseconds at laser intensity of 3.2×10^{14} W/cm². The deviation factor is actually larger than the present result. In this work, by taking into account of the Raman coupling with $4s$ states, we captured the main effects of NIR driving ionization and ionic coupling. In particular, the relative strength of the two absorption lines cannot be reproduced without the coherent Raman coupling. We would like to emphasize the reliability of our theoretical frame based on the ionization-coupling theory, which has been successfully validated in predicting the population of electronic states in the nitrogen ions induced by strong-field ionization [Communications Physics 3, 50 (2020), Nature Communications 13, 4080 (2022)]. Furthermore, we have expanded the theory to simulate the transient absorption spectra, and our simulation reproduces well the experimental results [PRL 129, 123002 (2022)] of nitrogen molecules (see <https://arxiv.org/abs/2310.04210>).

Table R1: Energy levels of Kr⁺. The energies of the corresponding states are calculated via Multi-Configuration Dirac-Fock method.

label	Configuration	J	Energy (eV)	Comments
0	[Ar]3d ¹⁰ 4s ² 4p ⁵	1.5	0.000	Kr ⁺ ground state
1	[Ar]3d ¹⁰ 4s4p ⁶	1.5	13.15	Coupling states considered in our calculations
2	[Ar]3d ¹⁰ 4s ² 4p ⁴ 4d	0.5	13.86	
3	[Ar]3d ¹⁰ 4s ² 4p ⁴ 5s	0.5	13.97	
4	[Ar]3d ¹⁰ 4s ² 4p ⁴ 5d	0.5	21.51	
5	[Ar]3d ¹⁰ 4s ² 4p ⁴ 5s	1.5	13.04	
6	[Ar]3d ¹⁰ 4s ² 4p ⁴ 4d	1.5	17.57	
7	[Ar]3d ¹⁰ 4s ² 4p ⁴ 5d	1.5	19.71	
8	[Ar]3d ¹⁰ 4s ² 4p ⁴ 5s	2.5	14.74	
9	[Ar]3d ¹⁰ 4s ² 4p ⁴ 4d	2.5	15.84	
10	[Ar]3d ¹⁰ 4s ² 4p ⁴ 5d	2.5	19.93	
11	[Ar]3d ¹⁰ 4s ² 4p ⁴ 6d	2.5	21.51	

We agree with you that more states can be involved. In order to corroborate our analysis, we have followed your advice and performed calculations with 11 essential multi-configuration states (see Table R1), and resulted in slightly increased time delays (240 as) for intensity I_1 , as shown in Fig. R1(d). The Raman effect can be pinpointed by looking into the instantaneous population and delay-dependent phase shift. It is known that the absorption profile is determined by oscillation strength and the phase shift of the dipole as described by the Eq. (17) in [Phys. Rev. A 86, 063411 (2012)]. The modulation of the absorption strength at the resonant energy can be written as $\rho_i \cos[\phi_i(\tau)]$. The phase shift can be retrieved from the simulation by taking the Fourier transform of the induced dipole, i.e. $\phi = \text{Arg}[d(\omega)]$ for the resonant transition energy ω_0 . In order to examine the origin of the phase shift, we calculate the instantaneous eigen energies by diagonalizing the instantaneous Hamiltonian of the 11 coupled states. Under this adiabatic picture, the phase shift can be approximated by $\varphi = \int \Delta E dt$, where ΔE is the adiabatic energy shift. It can be seen in Fig. R1 (a) that the phase variations with delay obtained by the two manners are qualitatively consistent. The discrepancy is more evident in $\cos[\phi_i(\tau)]$, as shown in Fig. R1(b). The adiabatic phase shift gives step-like rising, while the calculated phase gives half-cycle modulations arising from non-adiabatic effects. The population redistributed by the Raman coupling are plotted in Fig. R1(c). The modulations of the two population differ by relative time delay of about 40 attoseconds. As both the Raman coupling caused phase shift and population variation contribute to the resonant absorbance, the time delay of the two resonant absorption lines is found to be 240 attoseconds, as indicated in Fig. R1(d). Therefore, the multiple state calculation confirms the reliability of our model. We address this in the revised manuscript and include the 11-states calculation in the revised supplementary.

Fig.R1 Calculation results with 11 essential multi-configuration states (see Table R1) for the laser intensity I_1 . (a) The delay-dependent phase corresponding to the transitions $4p_{3/2}^{-1} \rightarrow 3d_{5/2}^{-1}$ (ϕ_1) and $4p_{1/2}^{-1} \rightarrow 3d_{3/2}^{-1}$ (ϕ_2) for the calculation (solid lines) and the adiabatic results (dot dash lines) ($\varphi_1: 4p_{3/2}^{-1} \rightarrow 3d_{5/2}^{-1}$ (red); $\varphi_2: 4p_{1/2}^{-1} \rightarrow 3d_{3/2}^{-1}$

¹(green)). (b) The cosine function corresponding to the calculated phase (solid lines) ϕ_1 (red) and ϕ_1 (green) and the adiabatic phase (dot dash lines) φ_1 (red) and φ_2 (green). (c) The time-dependent population of the two ionic states $4p_{3/2}^{-1}$ (red) and $4p_{1/2}^{-1}$ (green). (d) The absorption of the transitions $4p_{3/2}^{-1} \rightarrow 3d_{5/2}^{-1}$ (red) and $4p_{1/2}^{-1} \rightarrow 3d_{3/2}^{-1}$ (green) at their resonant energy in our calculation (solid lines) and the corresponding analytical results (dotted lines) obtained via $\rho_i \cos[\phi_i(\tau)]$.

Q3: Third, it is not clear if the simulations account for the nontrivial relation between population and absorption signal in the temporal region of pump-probe overlap; see Phys. Rev. A 86, 063411 (2012). Briefly, due to the finite lifetime of the Kr 3d core hole, the transient absorption probe also samples the time-dependent population after pump-probe time delay t_d . Hence, the transient absorption signal at pump-probe time delay t_d does not directly reflect the ion population at t_d .

R3:

We completely agree with you that the transient absorption signal at pump-probe time delay τ_d does not directly reflect the ion population at τ_d . Our calculations were performed with a procedure in accord with your point. In our calculation, we calculated the absorption spectra from the dipole response to the combined field of the NIR pulse and the attosecond pulse at given time delays, with the hole lifetime included. The spectral signal is naturally the integration of temporal response over all the time points. We have modified our manuscript by detailing the procedure of our calculations.

Q4: For the above reasons, in addition to the lack of novelty, I recommend publication of the manuscript in a more specialized physics journal. There, the authors could elaborate on the differences between the behaviors of Kr and Xe, expand their discussion of laser-coupling involving the 4s-1 states to include other ion excited states, address why they think that nonadiabatic effects do not explain the phase shift (as postulated by the authors in ref. 25), and propose other possible mechanisms that might give rise to the phase shift.

R4:

Thank you for the comments and suggestions, and we appreciate your re-evaluation. We hope our responses make the manuscript more convincing in terms of novelty and significance this time. We believe this work reveals new aspects of the coherent Raman effect and makes a solid advancement toward the complete understanding of attosecond absorptions by transiently ionized systems in strong laser field. On the other hand, as speculated in Ref. 25 and pointed out by you, the time-delay between the resonant absorptions can be affected by the ionization process, i.e, the coupling between the N-electron ground state and the monatomic ion plus excited electron states. In the present work, we have solved the ionization using strong field approximation which predicts correctly the overshoot of the ionization probability, partly taking into account the nonadiabatic ionization, the model however is limited to single active electron picture. Nevertheless, we believe the present work is so far much

more sophisticated and transparent, and will stimulate further investigations toward solving the strong field ionization of correlated systems.

REVIEWERS' COMMENTS

Reviewer #3 (Remarks to the Author):

I thank the authors for their reply. Like Reviewer #2, I am still unconvinced about the novelty of the work. For example, the authors claim that ref. 25 shows only the hole population dynamics whereas they show the absorption signal. (“In contrast, Ref. 25 discussed the time delay of the reconstructed density matrix rather than the absorption lines.”) Given that the relation between the two has previously been established, I would not consider the simulation of the absorption signal an advance sufficient to merit the publication of this work in *Nature Communications*. Furthermore, the authors claim that their ab initio method is more accurate (0.2 fs calculated vs. ~0.4 – 0.5 fs measured) than that employed in ref. 25 (75 as calculated vs. 200 as measured). This improvement, however, seems marginal. For the above reasons, I am hesitant to recommend publication of the manuscript in *Nature Communications*.

RESPONSES TO REVIEWERS

We appreciate the reviewer for his/her valuable time and effort spent on the re-evaluation of our work. Our detailed responses can be found below.

Reviewer #3:

I thank the authors for their reply. Like Reviewer #2, I am still unconvinced about the novelty of the work. For example, the authors claim that ref. 25 shows only the hole population dynamics whereas they show the absorption signal. (“In contrast, Ref. 25 discussed the time delay of the reconstructed density matrix rather than the absorption lines.”) Given that the relation between the two has previously been established, I would not consider the simulation of the absorption signal an advance sufficient to merit the publication of this work in Nature Communications. Furthermore, the authors claim that their ab initio method is more accurate (0.2 fs calculated vs. ~0.4 – 0.5 fs measured) than that employed in ref. 25 (75 as calculated vs. 200 as measured). This improvement, however, seems marginal. For the above reasons, I am hesitant to recommend publication of the manuscript in Nature Communications.

R1:

Thank you for the comments, and we appreciate your re-evaluation. We would like to emphasize again that **the novelty of this work lies in the effects of Raman coupling and the quantum coherence on the resonant absorptions**, which was not discussed in Ref. 25 or reported elsewhere. Either the absorption signals in our work or the hole population dynamics in Ref. 25 are just the experimental observations, the underneath physics revealed are much more valuable. The significance of Ref. 25 is the discovery of transient ground-state polarization that leads to reversible electronic population, while we report the coherent Raman coupling between the cation states that leads to extra time-delay between different transition channels. We think these are the fundamental and critical differences, which justify the novelty and noteworthiness of our manuscript. Regarding to theoretical method, the marginal improvement is not the key point. We think our model should not necessarily beat the TDCIS used in Ref. 25, but should provide new insights. The model developed in our manuscript is transparent, allowing us to capture the main effects of NIR driving ionization and ionic coupling.

We thank the reviewer again for his/her comments, and thank the editorial office for their instructions to improve the manuscript.

All the changes of the revised manuscript are marked in red, and the main changes are listed below:

- 1. The author affiliations and email address of one the corresponding error are updated;**
- 2. The structure of the manuscript was adjusted slightly by labelling each section;**
- 3. All the subfigures are relabeled without braces, and the corresponding references are revised accordingly.**
- 4. The error bar definition of Fig. 3 is added in the legend.**
- 5. All the references to supplementary information are updated with precise note number.**
- 6. The Data availability and Code availability are revised;**
- 7. An additional funding source is added;**
- 8. Typos and grammar errors are corrected.**